# Customizing Visual Emotion Evaluation for MLLMs: An Open-vocabulary, Multifaceted, and Scalable Approach

## Abstract

Recently, Multimodal Large Language Models (MLLMs) have achieved exceptional performance across diverse tasks, continually surpassing previous expectations regarding their capabilities. Nevertheless, their proficiency in perceiving emotions from images remains debated, with studies yielding divergent results in zero-shot scenarios. We argue that this inconsistency stems partly from constraints in existing evaluation methods, including the oversight of plausible responses, limited emotional taxonomies, neglect of extra-visual factors, and labor-intensive annotations. To facilitate customized visual emotion evaluation for MLLMs, we propose an Emotion Statement Judgment task that overcomes these constraints. Complementing this task, we devise an automated pipeline that efficiently constructs emotion-centric statements with minimal human effort. Through systematically evaluating prevailing MLLMs, our study showcases their stronger performance in emotion interpretation and context-based emotion judgment, while revealing relative limitations in direct determination of sentiment polarity and personalized emotion prediction. When compared to humans, even top-performing MLLMs like GPT-4o demonstrate remarkable performance gaps, underscoring key areas for future improvement. By developing a fundamental evaluation framework and conducting a comprehensive MLLM assessment, we hope this work contributes to advancing emotional intelligence in MLLMs. Codes and data will be released.

## 1 Introduction

Perceiving emotional signals from visual stimuli is essential for humans to improve decision-making and build effective communication [1, 2]. To computationally model this capability, Affective Image Content Analysis (AICA) has emerged as a key research direction in computer vision [3], focusing on emotion perception through visual features [4]. Over the decades, advances in this field have given rise to various applications, including opinion mining [5], customized advertising [6], and mental health care [7]. Recently, the advent of Multimodal Large Language Models (MLLMs) [8, 9] has revolutionized image understanding tasks [10]. However, their effectiveness in AICA remains contested. Divergent findings underscore a paradox: while some studies [11, 12] demonstrate MLLMs' poor zero-shot emotion recognition performance, others successfully employ them as emotion annotators for training data augmentation [13, 14]. We attribute this discrepancy to the partial incompatibility of conventional emotion evaluation approaches with MLLMs.

Specifically, current evaluation approaches can be broadly categorized into emotion classification and emotion interpretation, as illustrated in Fig. 1 (a,b). In emotion classification, models are required to assign the emotional state of an input image to a predefined set of emotion categories, with most

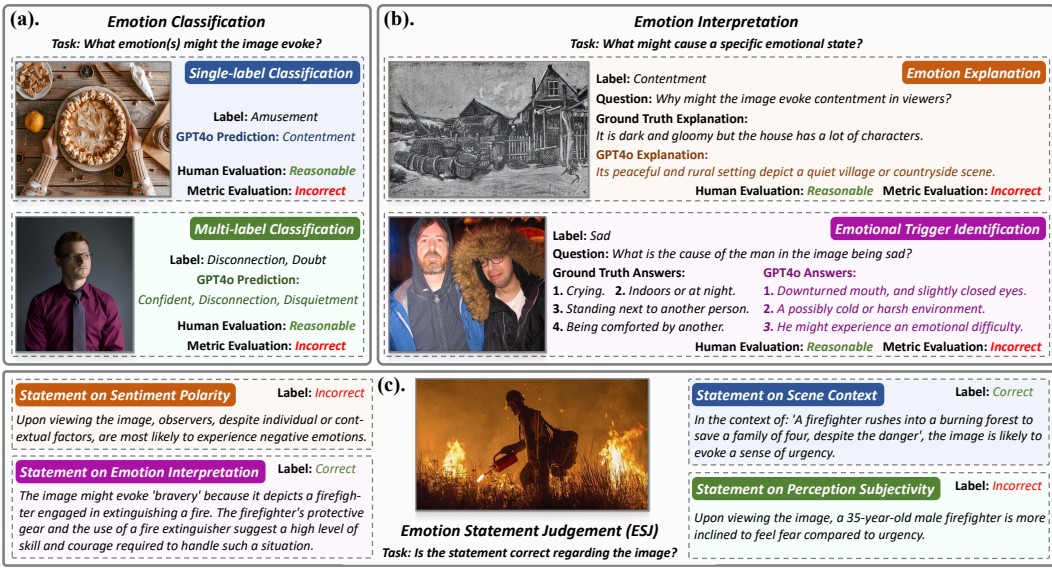

Figure 1: Illustration of different visual emotion evaluation approaches. Compared to current evaluation approaches, emotion statement judgement adopts a deterministic label while maintaining extensibility to evaluation depth and breadth.

benchmarks [15, 16, 17, 18, 19] providing a single label per image, and a few [20, 21] incorporating multiple labels. In contrast, emotion interpretation focuses on understanding the underlying causes of emotions in images. It encompasses two primary sub-tasks: explaining the causes of emotional states [22, 23] and identifying salient visual elements that contribute to the emotional response [24].

We identify four primary limitations when applying these evaluation approaches to MLLMs. *Firstly*, their adoption of fixed ground-truth answers for open-ended questions imposes structural constraints that exclude other plausible responses. Emotion perception is inherently subjective [25, 26], as the same image may evoke divergent reactions across individuals, and emotional states permit varied interpretations. As demonstrated in Fig. 1 (a,b), responses generated by GPT-4o that seem reasonable to humans are judged as inaccurate under rigid evaluation metrics. *Secondly*, they are mostly constructed upon emotion theories with limited emotional taxonomies. Popular emotion classification and interpretation benchmarks, such as FI [16] and Artemis [22], comprise only eight emotion categories. Such taxonomic granularity fails to capture fine-grained affective variations between images. *Thirdly*, they focus solely on intrinsic image attributes while neglecting extrinsic contextual factors. According to recent studies [4, 27], emotion perception can also be shaped by extra-visual factors, including the scene context where the image takes place, as well as the identity and personality of the viewer [25]. *Fourthly*, they predominantly rely on majority voting mechanisms to ensure label reliability in crowdsourced annotations [28], which is labor-intensive, particularly for fine-grained annotation tasks. EMOTIC [20], for instance, requires coordination with 23,788 annotators to label 18,316 images. This operational burden severely constrains dataset scalability in magnitude and generalization capacity across image domains.

To facilitate customized visual emotion evaluations for MLLMs, we propose a dual-component solution that addresses these limitations: the Emotion Statement Judgment (ESJ) task, complemented by the INSETS (**IN**telligent Vi**S**ual **E**motion **T**agger and **S**tatement Constructor) pipeline for efficient annotation. As a pioneering effort, we prioritize a precise over a complex evaluation design, aiming to establish a fundamental offline standard. With this purpose, **ESJ reformulates visual emotion evaluation by requiring MLLMs to validate emotion-centric statements for a given image**. It effectively mitigates ambiguity in open-ended questions while being highly extensible for evaluation depth and diversity. Meanwhile, INSETS annotates images with multiple open-vocabulary emotion labels, significantly refining the emotional taxonomies. These labels are then utilized to construct multifaceted emotion-centric statements, covering intrinsic image attributes like sentiment polarity and emotion interpretation, as well as extrinsic contextual factors like scene context and perception subjectivity. Crucially, only minimal human intervention is required, ensuring a high scalability of the approach. An example of ESJ is illustrated in Fig. 1 (c).

Leveraging INSETS, we introduce two ESJ benchmarks: a large-scale INSETS-462K and its human-refined subset INSETS-3K. Systematic evaluation demonstrates that recent MLLMs exhibit non-trivial visual emotion perception capabilities, yet maintain non-negligible performance gaps compared to humans, particularly in discerning sentiment polarity and comprehending perception subjectivity. In summary, the contributions of this paper are three-fold:

- This paper constitutes a pioneer effort to identify limitations in existing visual emotion evaluations for MLLMs and address them with a customized ESJ task.
- Complementing the ESJ task, this paper designs the INSETS pipeline, providing a reliable approach to annotating images with multiple open-vocabulary emotion labels and constructing multifaceted emotion-centric statements with minimal human effort.
- Utilizing the ESJ task and the INSETS pipeline, this paper conducts a systematic evaluation of recent MLLMs in visual emotion understanding, offering insights and fostering further developments of emotional intelligence in MLLMs.

## 2 Related Works

### 2.1 AICA Benchmarks

Psychological researchers conceptualize emotion representation through two principal frameworks: the Categorical Emotion Space (CES), which discretizes affective states into predefined taxonomies, and the Dimensional Emotion Space (DES), which maps emotions onto continuous 2D/3D coordinate systems (e.g., valence-arousal-dominance (VAD) axes [29]). For simplicity and better interpretability, most benchmarks adopt emotion classification evaluations based on discrete CES emotion taxonomies. This category encompasses both early small-scale benchmarks, such as IAPSa [30] and Abstract [15], as well as later larger-scale benchmarks like FI [16] and WebEmo [18]. Over time, benchmarks with enriched metadata have also been developed. Notable examples include EMOTIC [20], which integrates multiple emotion categories, VAD values, and human-related bounding boxes, and EmoSet [19], which employs describable emotion attributes that cover different levels of visual information.

Some other benchmarks adopt emotion interpretation evaluations by extending CES-based taxonomies with additional emotional explanations, such as Artemis [22] and Affection [23]. EIBench [24] diverges slightly, shifting focus on identifying and extracting visual emotional triggers. Based on these benchmarks, numerous expert models [31, 32, 33] have been developed, demonstrating strong performance under the fine-tuning and testing paradigm. In contrast to them, MLLMs are commonly pre-trained on web-scale data, without explicitly aligning with benchmark-specific knowledge. This discrepancy introduces multiple constraints when applying conventional benchmarks to MLLMs, necessitating customized visual emotion evaluation approaches that account for their generalized knowledge structures.

### 2.2 Evaluation of MLLMs

Recent years have witnessed surging academic and industrial interest in MLLMs. Unlike earlier models that are limited to specific domains, MLLMs demonstrate versatile competence across diverse tasks [34, 35], fueling expectations about their trajectory toward Artificial General Intelligence [36]. To evaluate MLLMs, various benchmarks have been established to examine their capabilities in areas such as perception [37, 9], reasoning [38, 39], ethics [40, 41], and specialized domains [42, 43]. However, emotional intelligence remains conspicuously underexplored. In existing efforts, FABA-Bench [44] evaluates MLLMs' comprehension of facial expressions and actions, MM-BigBench [45] simply aggregates mainstream image-text benchmarks, and EmoBench [46] is confined to solely language modality. To fill this gap, we propose the ESJ task and corresponding benchmarks INSETS-462K and INSETS-3K to advance more comprehensive visual emotion evaluation of MLLMs.

## 3 Emotion Statement Judgement

ESJ aims to evaluate the proficiency of MLLMs in perceiving emotions from visual content. In each evaluation trial, MLLMs receive an image and a paired emotion-centric statement. MLLMs are then required to judge the correctness of the statement in relation to the image by responding

with **Correct** or **Incorrect**. To ensure both depth and breadth in evaluation, we systematically design emotion-centric statements from four dimensions:

***Sentiment Polarity Statements*** require MLLMs to decide sentiment polarities without any additional clues. They assess MLLMs' proficiency in directly identifying the basic emotional tone.

***Emotion Interpretation Statements*** ask MLLMs to verify the consistency between affective explanations and corresponding emotional states. They measure MLLMs' affective reasoning capability given specific emotional triggers.

***Scene Context Statements*** probe MLLMs' comprehension of the dynamic interplay between the external scene context where the image takes place, and image-evoked emotional responses.

***Perception Subjectivity Statements*** task MLLMs to predict the personalized emotional responses under assumptions of specific viewer identities, examining whether MLLMs can recognize how subjectivity shapes emotional perceptions.

Collectively, these dimensions establish a holistic visual emotion evaluation framework for MLLMs. They cover both intrinsic image attributes emphasized in existing benchmarks and underexplored extrinsic contextual factors critical for human emotional perception [47, 48].

# 4    Annotation Pipeline: INSETS

Complementing the ESJ task, we design an automated pipeline for constructing emotion-centric statements, termed INSETS (**IN**telligent Vi**S**ual **E**motion **T**agger and **S**tatement Constructor). It operates through two primary stages: open-vocabulary emotion tagging and emotion statement construction. The prompts and templates used in the process are listed in Appendix Table 5.

## 4.1    Preliminary: Parrott's Hierarchical Model

We first introduce a well-established emotion model, which provides essential context for understanding the subsequent stages. Parrott's Hierarchical model [49, 50] is a tree-structured emotion taxonomy, comprising 6 primary emotions, 25 secondary emotions, and 113 tertiary emotions. The primary category contains three positive emotions (joy, love, and surprise) and three negative emotions (anger, fear, and sadness). Secondary emotions offer more diverse emotional states, each categorized under a corresponding primary emotion, while tertiary emotions further refine secondary emotions into more specific affective states. The complete taxonomy is presented in Appendix Table 7.

## 4.2    Open-vocabulary Emotion Tagging

At this stage, INSETS aims to assign open-vocabulary emotion labels for images, laying a solid foundation for constructing meaningful emotion-centric statements, with its procedure depicted in Fig. 2. According to Cheng *et al.* [14], MLLMs demonstrate promising capabilities in generating emotional descriptions from visual content and extracting underlying emotions from these descriptions. However, challenges such as hallucinations [51], trustworthiness issues [52], and inherent limitations in emotional perception can lead to inaccuracies in the extracted emotions. To enhance reliability, we devise an ensemble-based majority voting mechanism, aggregating outputs from multiple MLLMs to cross-validate and refine emotion label assignments.

Given an image sample, we first extract its potential open-vocabulary emotions from multiple MLLMs. MLLMs are prompted to analyze the emotions evoked by the image (with #1 prompt in Table 5, abbreviated as "#1" in the following) and then extract emotions applicable to the image (#2) [Fig. 2 (a)]. This process is iteratively applied to all images in the dataset, aggregating potential emotions into an emotion pool. Next, we refine this pool by filtering out words unsuitable as emotion descriptors (#3), using GPT-4 [53] as the judge due to its superior linguistic emotional perception [46] [Fig. 2 (b)]. Once the filtered emotion pool is obtained, we attach the remaining emotions to Parrott's hierarchical emotion model [Fig. 2 (c)]. Specifically, GPT-4 is prompted to categorize each open-vocabulary emotion into the closest tertiary emotion in Parrott's model (#4), followed by manual refinement by a human expert. This process results in an extended version of Parrott's model, which we refer to as the Parrott-based Open-vocabulary Hierarchical Model (POM) [Fig. 2 (d)]. This unified framework

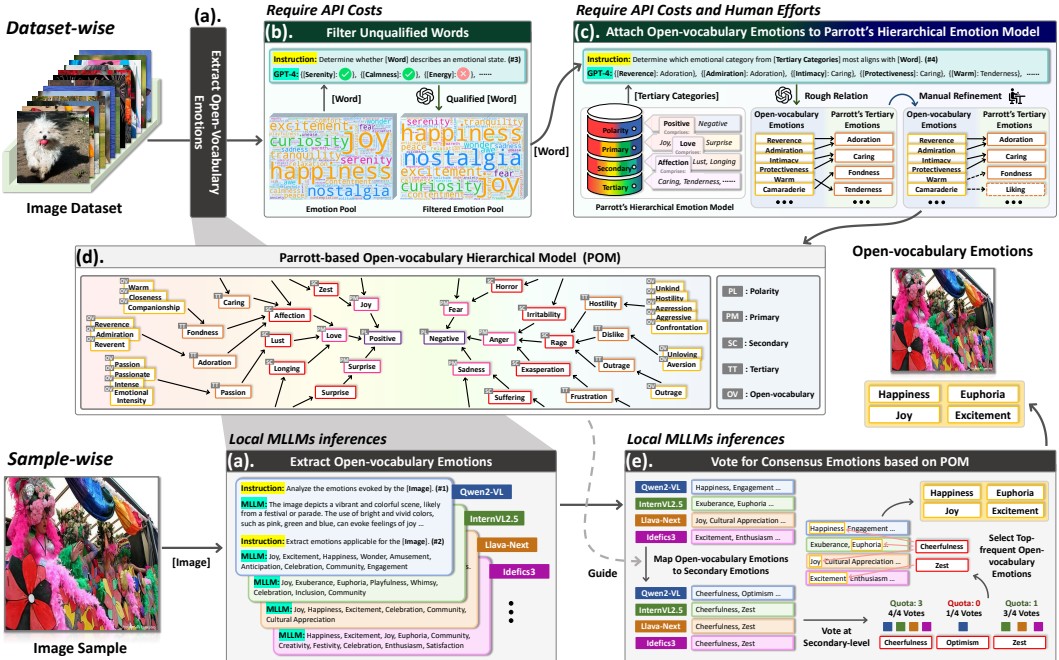

Figure 2: Illustration of the open-vocabulary emotion tagging stage. We first extract all potential open-vocabulary emotions from the image dataset (a) and then attach these emotions to a well-established emotion model (b,c). Through this model (d), we identify and select open-vocabulary emotions consistently recognized by multiple MLLMs as the labels of each image (e).

enables multi-level tracing of affective states for each open-vocabulary emotion, facilitating more accurate and interpretable emotion tagging.

Subsequently, leveraging POM, the ensemble-based majority voting mechanism selects reliable open-vocabulary emotion labels for images [Fig. 2 (e)]. First, open-vocabulary emotions extracted from multiple MLLMs are mapped to secondary emotion categories, with model voting determining the quota of open-vocabulary emotions for each secondary category. Second, within each secondary category, the open-vocabulary candidates are ranked based on their frequencies, and the top-ranked ones are selected according to the allocated quota. This process guarantees label reliability while preserving the open-vocabulary nature of the emotion labels, thereby achieving synergistic optimization between annotation precision and semantic coverage.

## 4.3 Emotional Statement Construction

Building upon the assigned emotion labels, we construct both correct and incorrect emotion-centric statements from four dimensions: sentiment polarity, emotion interpretation, scene context, and perception subjectivity. Its procedure is illustrated in Fig. 3.

The construction pipeline initiates with prototype statement generation [Fig. 3 (a)]. For each emotion label, we trace it back to the MLLM that extracts it, prompting the MLLM to generate three prototype statements: **[1]**. *prototype interpretation* of the emotion by inquiring about the cause of the emotion (#5); **[2]**. *prototype context* that aligns with the emotion by requesting a background story (#6); and **[3]**. *prototype character* who would experience the emotion by questioning the possible identity of the viewer (#7). From the dataset perspective, the prototype generation is distributed across multiple MLLMs, ensuring diversity in the subsequent statement construction.

***Sentiment Polarity Statement Construction*** [Fig. 3 (b)]: Under the guidance of POM, we derive sentiment polarity by mapping the labels to primary emotions. Each image's sentiment polarity is classified into three mutually exclusive categories: **1). Fully Positive** when all labels reside in the positive spectrum; **2). Fully Negative** when all labels reside in the negative spectrum; **3). Mixed** when positive and negative labels both exist. Next, the ground truth correctness of three predefined statements on sentiment polarity (#8,9,10) is determined accordingly.

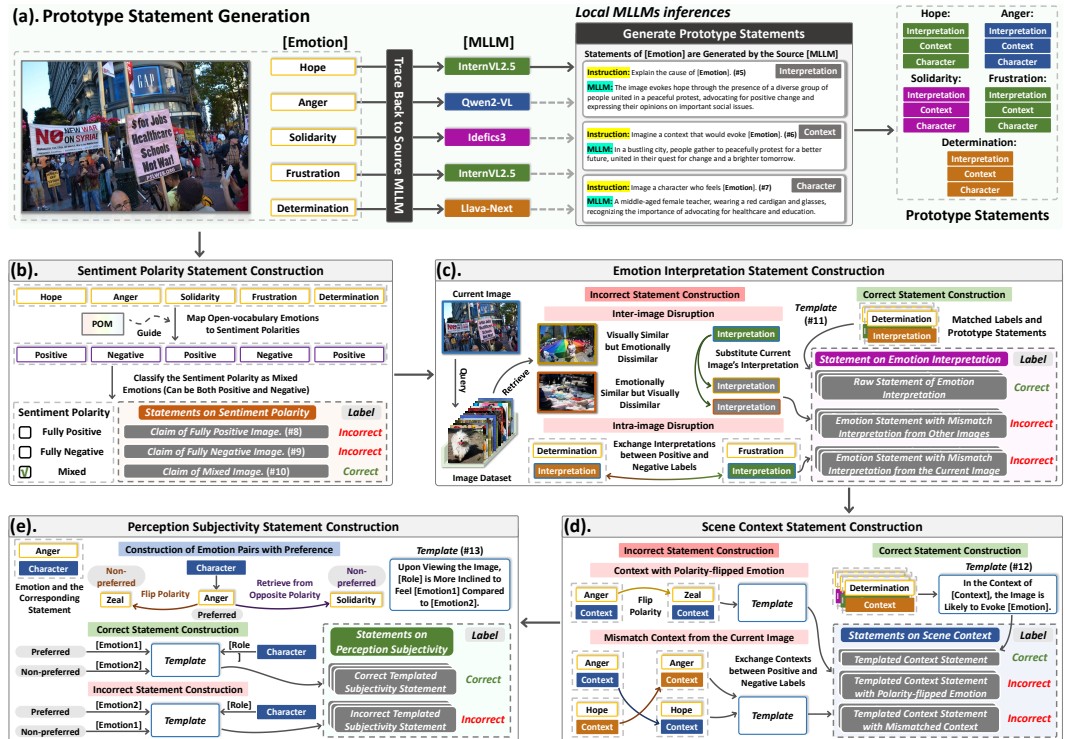

Figure 3: Illustration of the emotional statement construction stage. It begins with prototype statement generation (a) for each emotion label, which is distributed across multiple MLLMs. Then, based on the assigned emotion labels and the corresponding prototype statements, correct and incorrect emotion-centric statements are constructed from four dimensions: sentiment polarity (b), emotion interpretation (c), scene context (d), and perception subjectivity (e).

***Emotion Interpretation Statement Construction*** [Fig. 3 (c)]: We combine a prototype interpretation with an emotional state (#11) to construct an emotion interpretation statement. Matched labels and prototype statements are assigned as correct statements, while unmatched ones are considered incorrect. To construct mismatched pairs, we design two strategies: inter-image and intra-image disruption. The former retrieves two images from the dataset - one exhibiting visual similarity but emotional dissimilarity to test whether MLLMs can comprehend the affective gap [54], and the other demonstrating emotional similarity but visual dissimilarity to evaluate whether MLLMs can identify the emotional triggers in images. Visual similarity is quantified using CLIP-score [55], whereas emotional similarity is determined by the correspondence at tertiary emotions of POM. The latter strategy exchanges interpretations between emotion labels of contrasting polarity within identical images, aiming to assess whether MLLMs can establish precise causal linkages between emotional triggers and specific emotions.

***Scene Context Statement Construction*** [Fig. 3 (d)]: We combine a prototype context and an emotional conclusion (#12) to form a scene context statement, where the construction of correct statements mirrors the previous case. The strategy for incorrect statements differs slightly: in addition to exchanging prototype contexts between emotion labels of contrasting polarity within identical images, we devise a flip polarity operation. Specifically, the emotional label is substituted with a tertiary emotion randomly sampled from the opposing polarity spectrum in POM.

***Perception Subjectivity Statement Construction*** [Fig. 3 (e)]: We combine a prototype character and their inclination toward one of two candidate emotions (#13) to form a perception subjectivity statement. We first create emotion pairs with preference bias. For each prototype character, its preferred emotion is the corresponding emotion label. The non-preferred emotion is derived by either retrieving opposite polarity emotions within the image or the flip polarity operation for the preferred emotion. Next, correct and incorrect statements are constructed by configuring emotion pairs into canonical or anomalous orders, respectively.

Table 1: Statistics of the MLLMs employed in INSETS. For each MLLM, we report the number of parameters, the average extracted emotions per image, the number selected as emotion labels, and the proportion of prototype statements it generates.

| MLLMs | #P (B) | Extracted Emotion | Selected Emotion | Generated Statement |
|---|---|---|---|---|
| LLaVa-1.6 [56] | 7.6 | 8.3 | 2.4 | 9.8% |
| Mantis [57] | 8.5 | 12.6 | 2.9 | 13.1% |
| mPLUG-Owl3 [58] | 8.1 | 9.2 | 2.7 | 11.2% |
| Idefics3 [59] | 8.5 | 10.0 | 2.9 | 12.5% |
| Phi-3.5-Vision [60] | 4.1 | 9.9 | 2.8 | 11.7% |
| Qwen2-VL [61] | 8.3 | 8.8 | 2.7 | 10.9% |
| Llama-3.2-Vision [62] | 10.7 | 7.2 | 2.3 | 9.3% |
| Molmo [63] | 8.0 | 10.8 | 2.7 | 12.0% |
| InternVL2.5 [64] | 8.3 | 8.5 | 2.3 | 9.5% |

Table 2: Statistics of emotion labels and statements in INSETS-462K and INSETS-3K.

| INSETS-462K | |
|---|---|
| Number of Images | 17,716 |
| Number of Statements | 462,369 |
| Emotion Labels Per Image | 4.9 |
| Distinct Emotion Labels | 751 |
| Statements Per Image | 26.1 |
| Average Length of Statements | 39.0 |

| INSETS-3K | |
|---|---|
| Number of Images | 3,086 |
| Number of Statements | 3,086 |
| Emotion Labels Per Image | 5.2 |
| Distinct Emotion Labels | 424 |
| Statements Per Image | 1.0 |
| Average Length of Statements | 37.0 |

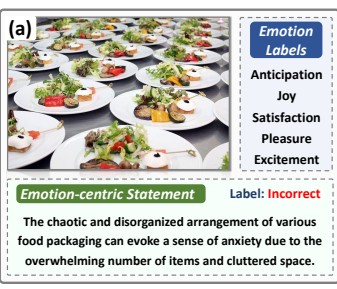
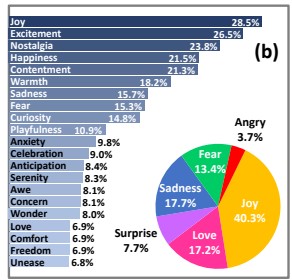
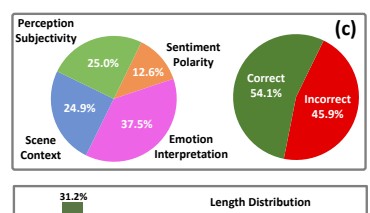
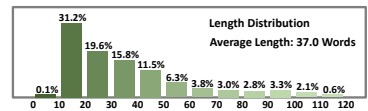

Figure 4: A closer gaze at INSETS-3K. Illustrations of a sample (a), the distribution of emotion labels (b), and the distribution of emotion-centric statements (c).

## 4.4 Evaluation Benchmarks

In this process, INSETS begins with an image dataset, relying primarily on local MLLM inferences to assign reliable open-vocabulary emotion labels and construct meaningful emotion-centric statements. For open-vocabulary emotion extraction and prototype statement generation, we employ nine recent popular MLLMs with impressive performance [65]. As reported in Table 1, the final assigned emotion labels and prototype statements are evenly sourced across the MLLMs, ensuring diversity in the constructed benchmark. Given the high quality of Emoset [19], we select it as the image source. From a subset of 17,716 images, we construct 462K emotion-centric statements, referred to as INSETS-462K.

To further enhance accuracy, we sample 3,164 distinct image-statement pairs from INSETS-462K for human validation. Among them, annotators judge 218 pairs (6.9%) as inaccurate, 2,868 pairs (90.6%) as accurate, and 79 pairs (2.5%) as ambiguous. Accurate pairs are retained, incorrect labels are corrected, and ambiguous ones are discarded, yielding a high-quality subset of 3,086 distinct image-statement pairs, which we name INSETS-3K. Although the open-vocabulary emotion labels are not required in the ESJ task, we retain them in both INSETS-462K and INSETS-3K to enhance interpretability and facilitate future development of the benchmarks. The statistics of emotion labels and statements are reported in Table 2.

## 5 Analysis and Evaluation

### 5.1 Details of INSETS-3K

To gain deeper insights into the properties of INSETS-3K, we provide detailed statistics in Fig. 4. A sample is shown in Fig. 4 (a), which includes five emotion labels and an emotion-centric statement. Fig. 4 (b) illustrates the distribution of popular emotion labels, where the most frequent labels include Joy, Excitement, Nostalgia, Happiness, and Contentment. When mapped to the primary emotions in Parrott's model, Joy is the most dominant category (40.3%), followed by Sadness (17.7%),

Table 3: Evaluation results on INSETS-3K. The MLLMs involved in constructing INSETS-3K are listed in the upper part of the table, while other results are listed in the lower part. The highest values in each section are marked in **bold**.

| MLLMs | #Param | Accuracy | | | | | Positive Ratio | Give-up Ratio |
|---|---|---|---|---|---|---|---|---|
| | | Sentiment Polarity | Emotion Interpretation | Scene Context | Perception Subjectivity | Total | | |
| LLaVa-1.6 [56] | 7.6B | 66.4 | 69.7 | 55.3 | 49.7 | 60.2 | 18.4 | 0 |
| Mantis [57] | 8.5B | 61.2 | 65.9 | 67.2 | 61.2 | 64.4 | 84.4 | 0.1 |
| mPLUG-Owl3 [58] | 8.1B | 73.9 | 79.3 | 81.7 | **75.0** | **78.1** | 67.3 | 0 |
| Idefics3 [59] | 8.5B | 75.4 | 78.6 | 75.5 | 62.6 | 73.4 | 49.5 | 0.2 |
| Phi-3.5-Vision [60] | 4.1B | 74.7 | 72.5 | 82.6 | 74.8 | 75.9 | 64.1 | 0 |
| Qwen2-VL [61] | 8.3B | 70.7 | 75.0 | **86.1** | 72.8 | 76.6 | 65.7 | 0 |
| Llama-3.2-Vision [62] | 10.7B | 68.7 | 75.9 | 85.2 | 72.0 | 76.3 | 71.2 | 0.2 |
| Molmo [63] | 8.0B | 61.4 | 76.0 | 79.2 | 59.4 | 70.7 | 38.1 | 0 |
| InternVL2.5 [64] | 8.3B | **75.7** | **80.2** | 79.4 | 61.3 | 74.7 | 52.9 | 0.2 |
| BLIP2 [66] | 7.7B | 51.1 | 52.8 | 55.4 | 52.5 | 53.2 | 96.8 | 2.5 |
| InstructBLIP [67] | 7.9B | 29.8 | 40.5 | 33.9 | 37.8 | 36.8 | 43.8 | 37.5 |
| Otter [68] | 8.2B | 32.6 | 21.4 | 32.1 | 27.2 | 27.0 | 9.9 | 52.1 |
| DeepSeek-VL [69] | 7.3B | 68.7 | 70.8 | 81.1 | **73.2** | 73.7 | 73.1 | 0 |
| Paligemma [70] | 2.9B | 50.6 | 46.3 | 49.3 | 45.7 | 47.4 | 49.4 | 5.5 |
| MiniCPM [71] | 8.7B | 70.4 | 78.4 | 81.9 | 70.5 | 76.2 | 66.0 | 0 |
| Qwen2.5-VL [72] | 8.3B | 63.2 | 81.5 | **83.9** | 66.3 | 75.9 | 45.9 | 0 |
| GPT4o-mini [53] | – | 62.5 | 80.0 | 78.9 | 71.8 | 75.4 | 49.5 | 0 |
| GPT4o [53] | – | **72.5** | **84.3** | 81.6 | 69.2 | **78.3** | 65.0 | 1.6 |

Love (17.2%), Fear (13.4%), Surprise (7.7%), and Anger (3.7%). This distribution suggests a rich representation of emotions, ensuring coverage of diverse affective states. Fig. 4 (c) presents statistics on the statements, which exhibit a natural length distribution and a well-balanced distribution across the four evaluation dimensions as well as correct/incorrect labels.

## 5.2 Evaluation Preparations

We evaluate MLLMs through the ESJ task. Specifically, we provide each MLLM with an image-statement pair and prompt it to determine the correctness of the statement. The prompt is formulated as: *"Based on the provided image and emotional statement, please determine whether the statement aligns with the content of the image. If it does, respond with **Correct**. If it does not, respond with **Incorrect**."* Each image-statement pair is queried three times per MLLM, and the most frequent response is selected as the final decision. After collecting responses for all images, we adopt accuracy as the primary evaluation metric. As identified in prior work [73], some MLLMs may exhibit a strong bias toward either positive or negative responses, which may compromise accuracy-based evaluation validity. To mitigate this, we introduce two diagnostic metrics: *Positive Ratio* calculates the proportion of positive responses out of all responses; *Give-up Ratio* measures the proportion of cases where the MLLM neither provides a positive nor a negative response.

To conduct a comprehensive evaluation, we adopt a diverse range of MLLMs, including both open-source and closed-source ones. Besides the MLLMs used to construct the benchmarks, we also incorporate the following MLLMs: BLIP-2 [66], InstructBLIP [67], Otter [68], Deepseek-VL [69], Paligemma [70], MiniCPM [71], Qwen2.5-VL [72], GPT4o-mini, and GPT4o [53]. All experiments are conducted on NVIDIA GeForce RTX 4090 GPUs.

## 5.3 Results and Findings

The evaluation results on INSETS-3K are reported in Table 3. Overall, more recent MLLMs outperform earlier MLLMs. The latter suffers from severe response biases or instructional failures. This indicates that advancements in general visual tasks also enhance emotional perception. However, no MLLM achieves optimal performance across all tasks. While InternVL2.5 and GPT4o demonstrate superior performance in identifying the basic emotional tone and performing affective reasoning, they exhibit comparative deficiencies in contextual and personalized emotion prediction. This underscores the multifaceted challenges of visual emotion understanding. Further evaluation on INSETS-462K is reported in Appendix Table 6.

Table 4: Evaluation results of MLLMs and humans on a subset of INSETS-3K containing 300 image-statement pairs.

| MLLMs | #Param | Accuracy | | | | | Positive Ratio | Give-up Ratio |
|---|---|---|---|---|---|---|---|---|
| | | Sentiment Polarity | Emotion Interpretation | Scene Context | Perception Subjectivity | Total | | |
| mPLUG-Owl3 [58] | 8.1B | 74.6 | **80.4** | 82.9 | **77.2** | **79.5** | 67.3 | 0 |
| Phi-3.5-Vision [60] | 4.1B | 75.4 | 72.9 | **83.9** | 73.3 | 76.1 | 64.0 | 0 |
| InternVL2.5 [64] | 8.1B | **77.2** | 79.5 | 79.3 | 63.2 | 75.1 | 52.1 | 0 |
| DeepSeek-VL [69] | 7.3B | 70.2 | 70.5 | 80.2 | **73.7** | 73.7 | 73.5 | 0 |
| MiniCPM [71] | 8.7B | 70.2 | 78.9 | 82.4 | 72.4 | 77.1 | 65.4 | 0 |
| Qwen2.5-VL [72] | 8.3B | 64.0 | 81.5 | **83.3** | 68.0 | 76.4 | 47.4 | 0 |
| GPT4o-mini [53] | - | 64.0 | 79.2 | 77.5 | 71.3 | 74.9 | 49.8 | 0 |
| GPT4o [53] | - | **73.7** | **84.5** | 81.2 | 71.1 | **79.0** | 64.6 | 0.6 |
| Human Average | – | 92.3 | 90.1 | 95.3 | 89.6 | 91.6 | 53.4 | 0 |
| Human Best | – | 97.4 | 95.8 | 98.7 | 94.7 | 95.2 | – | – |

***Comparison with Human Performance***: We sample 300 image-statement pairs from INSETS-3K and evaluate 25 human participants alongside leading MLLMs. As shown in Table 4, humans achieve near-perfect accuracy. In contrast, MLLMs exhibit notable performance gaps, particularly in determining sentiment polarity and understanding perception subjectivity. Given their comparatively high accuracy on emotion interpretation statements, we suggest that the affective reasoning from emotional clues to emotional states is essential for MLLMs to perceive emotions. Regarding perception subjectivity, we speculate that MLLMs may lack sufficient awareness of individual differences. Overall, ESJ is a fundamental task format, and the performance gap between MLLMs and humans highlights the considerable potential for improvement in MLLMs' visual emotional intelligence.

# 6   Limitations and Discussion

Several limitations in this work can be further improved. *First*, our evaluation primarily focuses on MLLMs with parameters under 10B due to computational constraints imposed by hardware. Although this covers practical deployment scenarios, it excludes larger-scale open-source MLLMs that may exhibit superior visual emotion perception capabilities. *Second*, the current implementation is limited to monolingual evaluation. Yet we highlight that adapting INSETS for multilingual construction would require relatively limited engineering effort, primarily involving adjustments in MLLM selection, prompt design, and template configuration. Moreover, while we explored basic reasoning strategies, advanced strategies such as in-context learning and chain-of-thought prompting remained underexplored. Systematically investigating these techniques can potentially reveal deeper insights into MLLMs' visual emotion perception mechanisms. *Third*, due to the lack of human validation, the INSETS-462K benchmarks inevitably incorporate certain noises and inaccuracies. Nevertheless, the validation process of the INSETS-3K benchmark suggests that over 85% image-statement pairs retain high accuracy. This can be attributed to our efforts in ensuring reliability, including the ensemble-based majority voting mechanism and necessary human interventions.

Future work includes expanding MLLM coverage, extending INSETS to support multilingual construction and further refinement of the human-AI collaborative annotation process.

# 7   Conclusion

In this paper, we propose the ESJ task and a complemented automated pipeline, INSETS, to advance open-vocabulary, multifaceted, and scalable visual emotion evaluation in MLLMs. Through them, we provide a nuanced and comprehensive evaluation framework that covers sentiment polarity, emotion interpretation, scene context, and perception subjectivity. Our evaluation reveals that while MLLMs exhibit certain capabilities in visual emotion perception, they still lag behind humans, highlighting the necessity of developing emotion-oriented training objectives. We hope that INSETS-3K and INSETS-462K can serve as reliable benchmarks to advance future research, fostering the development of MLLMs with improved emotional reasoning and understanding.

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

# A    Links of MLLMs

We provide the links to the model cards of the MLLMs we evaluated in the experiments.

LLaVa-1.6 [56]

`https://huggingface.co/llava-hf/llava-v1.6-mistral-7b-hf`

Mantis [57]

`https://huggingface.co/TIGER-Lab/Mantis-8B-siglip-llama3`

mPLUG-Owl3 [58]

`https://huggingface.co/mPLUG/mPLUG-Owl3-7B-241101`

Idefics3 [59]

`https://huggingface.co/HuggingFaceM4/Idefics3-8B-Llama3`

Phi-3.5-Vision [60]

`https://huggingface.co/microsoft/Phi-3.5-vision-instruct`

Qwen2-VL [61]

`https://huggingface.co/Qwen/Qwen2-VL-7B-Instruct`

Llama-3.2-Vision [62]

`https://huggingface.co/meta-llama/Llama-3.2-11B-Vision-Instruct`

Molmo [63]

`https://huggingface.co/allenai/Molmo-7B-D-0924`

InternVL2.5 [64]

`https://huggingface.co/OpenGVLab/InternVL2_5-8B`

BLIP2 [66]

`https://huggingface.co/Salesforce/blip2-opt-6.7b-coco`

InstructBLIP [67]

`https://huggingface.co/Salesforce/instructblip-vicuna-7b`

Otter [68]

`https://huggingface.co/luodian/OTTER-Image-LLaMA7B-LA-InContext`

DeepSeek-VL [69]

`https://huggingface.co/deepseek-ai/deepseek-vl-7b-chat`

Paligemma [70]

`https://huggingface.co/google/paligemma-3b-pt-448`

MiniCPM [71]

`https://huggingface.co/openbmb/MiniCPM-o-2_6`

Qwen2.5-VL [72]

`https://huggingface.co/Qwen/Qwen2.5-VL-7B-Instruct`

# B    Prompts and Statement Templates

The prompts and statement templates used in the INSETS pipeline are presented in Table 5.

Table 5: Prompts and statement templates employed in the INSETS pipeline.

| | Prompts and Statement Templates |
|---|---|
| #1 | You are an Emotional Perception Expert. Please analyze the emotions that might be evoked by the given image. Your analysis should explore a wide range of visual attributes, such as brightness, colorfulness, depicted scenes, objects, human actions, and facial expressions. Additionally, provide detailed explanations linking these attributes to the emotions they may trigger. If applicable, discuss any potential cultural or psychological factors influencing these emotional responses. |
| #2 | You are an Emotional Perception Expert. Your task is to extract all applicable emotions as comprehensively as possible based on the image description. Focus on distinct emotions such as happiness, sadness, fear, anger, etc. Keep the list concise, with a maximum of 10 distinct emotions. |
| #3 | You are tasked with determining whether the word "**[word]**" describes a specific emotional state. An emotional state is a psychological condition involving feelings and reactions triggered by internal or external events. Respond with "Yes" if the word aligns with this definition, or "No" otherwise. The output format should be {"word": "response"}. |
| #4 | You are tasked with assigning the word "**[word]**" to the most closely related emotional category from the following 115 predefined options: "**[categories]**". Consider broader semantic connections and possible emotional nuances when making your judgment. If the word cannot reasonably fit any category, respond with "not applicable". Do not create or assign new categories outside of the provided list. Do not provide any explanations or reasons for your choice. The output format should be {"word": "response"}. |
| #5 | Briefly explain why this image might evoke "**[emotion]**" in viewers, without mentioning any other emotions. |
| #6 | Imagine a background story for the image that would evoke a sense of "**[emotion]**" in viewers. Respond in one sentence. Do not mention the content in the image. |
| #7 | Imagine a character who would feel "**[emotion]**" when viewing this image. Include details such as their age, gender, profession, and other relevant traits. Describe the character in one concise sentence without further explanation. |
| #8 | Upon viewing this image, observers, despite various individual or contextual factors, are most likely to experience positive emotions. |
| #9 | Upon viewing this image, observers, despite various individual or contextual factors, are most likely to experience negative emotions. |
| #10 | Upon viewing this image, observers are equally likely to experience either positive or negative emotions, depending on individual or contextual factors. |
| #11 | Therefore, the image might evoke "**[emotion]**" in viewers. |
| #12 | In the context of: "**[context]**", the image is likely to evoke a sense of "**[emotion]**". |
| #13 | Upon viewing the image, "**[role]**" is more inclined to feel "**[emotion1]**" compared to "**[emotion2]**". |

Table 6: Evaluation results of MLLMs on INSETS-462K. (SP: Sentiment Polarity, EI: Emotion Interpretation, SC: Scene Context, PS: Perception Subjectivity)

| MLLMs | #P | Accuracy | | | | |
|---|---|---|---|---|---|---|
| | (B) | SP | EI | SC | PS | Total |
| mPLUG-Owl3 [58] | 8.1 | 64.3 | **78.1** | 80.5 | **78.3** | **77.4** |
| Phi-3.5-Vision [60] | 4.1 | 66.7 | 63.7 | 82.2 | 76.6 | 70.3 |
| InternVL2.5 [64] | 8.1 | **71.4** | 69.9 | **82.8** | 75.0 | 73.5 |
| DeepSeek-VL [69] | 8.7 | 58.3 | 60.2 | 86.2 | 77.0 | 68.7 |
| MiniCPM [71] | 8.7 | **64.3** | 77.1 | **86.8** | **83.6** | **79.3** |
| Qwen2.5-VL [72] | 8.3 | 58.3 | **81.3** | 76.4 | 63.9 | 74.3 |

## C  Details of Parrott's Hierarchical Model

We present the complete emotion taxonomy of Parrott's hierarchical model in Table 7.

## D  Further evaluation on INSETS-462K

Despite dataset noise, obvious task-specific disparities persist: MiniCPM surpasses Qwen2.5-VL in judging scene context statements (86.8% vs. 76.4%) but trails in judging emotion interpretation statements (77.1% vs. 81.3%). This finding reinforces the need for emotion-oriented training objectives in the future MLLM developments.

Table 7: Emotion taxonomy of Parrott's hierarchical model.

| Primary Emotion | Secondary Emotion | Tertiary Emotion |
|---|---|---|
| Love | Affection | Adoration, Fondness, Liking, Attraction, Caring, Tenderness, Compassion, Sentimentality |
| | Lust | Desire, Passion, Infatuation |
| | Longing | Longing |
| Joy | Cheerfulness | Amusement, Bliss, Gaiety, Glee, Jolliness, Joviality, Joy, Delight, Enjoyment, Gladness, Happiness, Jubilation, Elation, Satisfaction, Ecstasy, Euphoria |
| | Zest | Enthusiasm, Zeal, Excitement, Thrill, Exhilaration |
| | Contentment | Pleasure |
| | Pride | Triumph |
| | Optimism | Eagerness, Hope |
| | Enthrallment | Enthrallment, Rapture |
| | Relief | Relief |
| Surprise | Surprise | Amazement, Astonishment |
| Anger | Irritability | Aggravation, Agitation, Annoyance, Grouchy, Grumpy, Crosspatch |
| | Exasperation | Frustration |
| | Rage | Anger, Outrage, Fury, Wrath, Hostility, Ferocity, Bitterness, Hatred, Scorn, Spite, Vengefulness, Dislike, Resentment |
| | Disgust | Revulsion, Contempt, Loathing |
| | Envy | Jealousy |
| | Torment | Torment |
| Sadness | Suffering | Agony, Anguish, Hurt |
| | Sadness | Depression, Despair, Gloom, Glumness, Unhappiness, Grief, Sorrow, Woe, Misery, Melancholy |
| | Disappointment | Dismay, Displeasure |
| | Shame | Guilt, Regret, Remorse |
| | Neglect | Alienation, Defeatism, Dejection, Embarrassment, Homesickness, Humiliation, Insecurity, Insult, Isolation, Loneliness, Rejection |
| | Sympathy | Pity, Mono no aware, Sympathy |
| Fear | Horror | Alarm, Shock, Fear, Fright, Horror, Terror, Panic, Hysteria, Mortification |
| | Nervousness | Anxiety, Suspense, Uneasiness, Apprehension, Worry, Distress, Dread |

# E    Visualization of INSETS-3K

More samples from INSETS-3K are visualized in Fig. 5, Fig. 6, Fig. 7, Fig. 8, Fig. 9, Fig. 10, Fig. 11, Fig. 12.

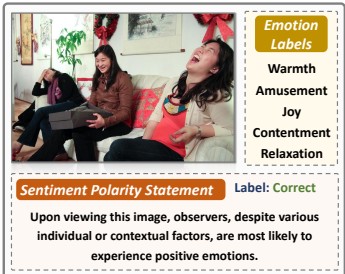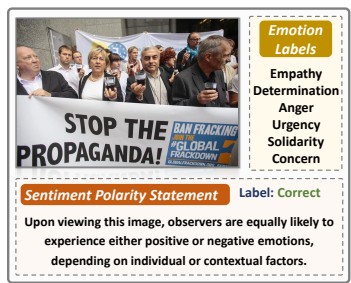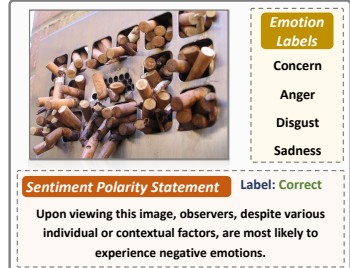

Figure 5: Correct sentiment polarity statements.

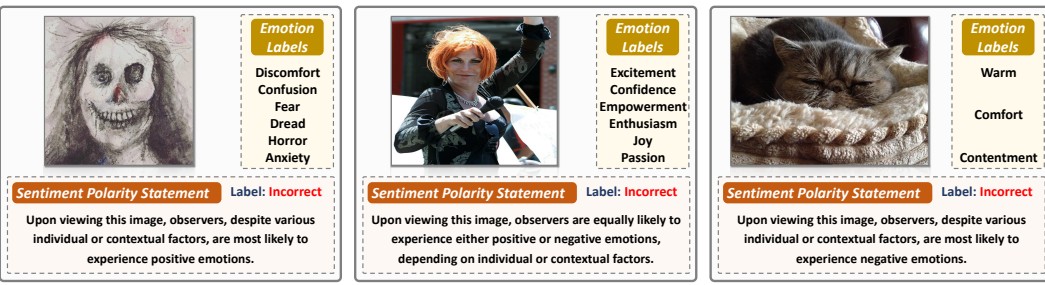

Figure 6: Incorrect sentiment polarity statements.

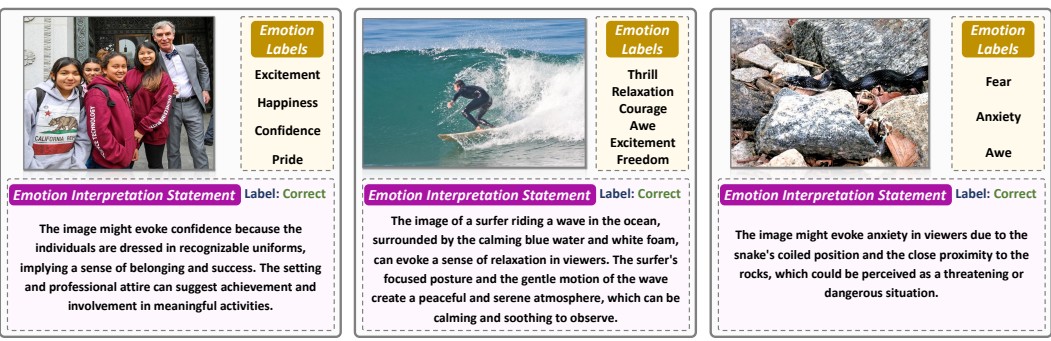

Figure 7: Correct emotion interpretation statements.

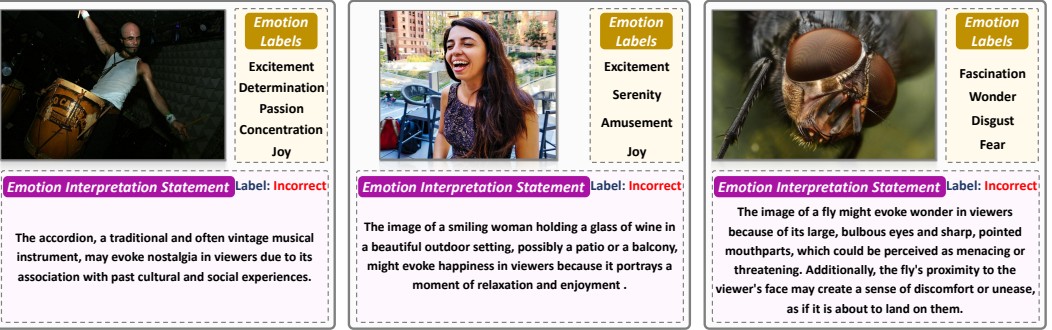

Figure 8: Incorrect emotion interpretation statements.

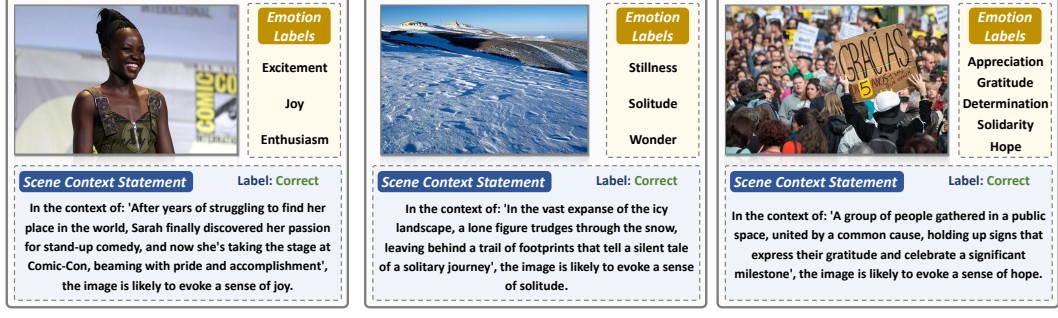

Figure 9: Correct scene context statements.

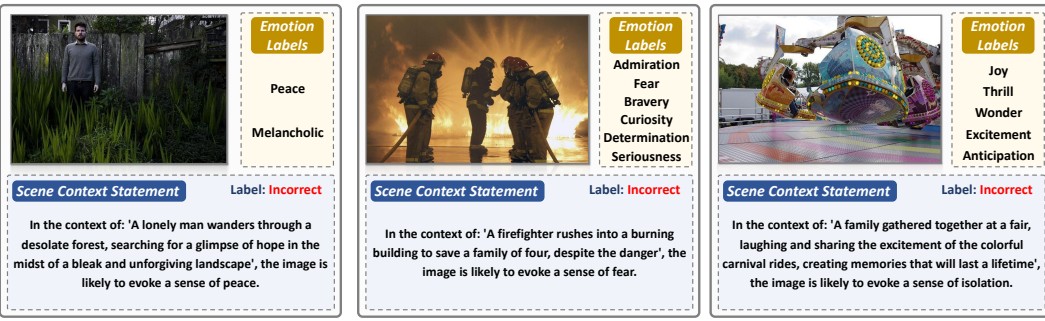

Figure 10: Incorrect scene context statements.

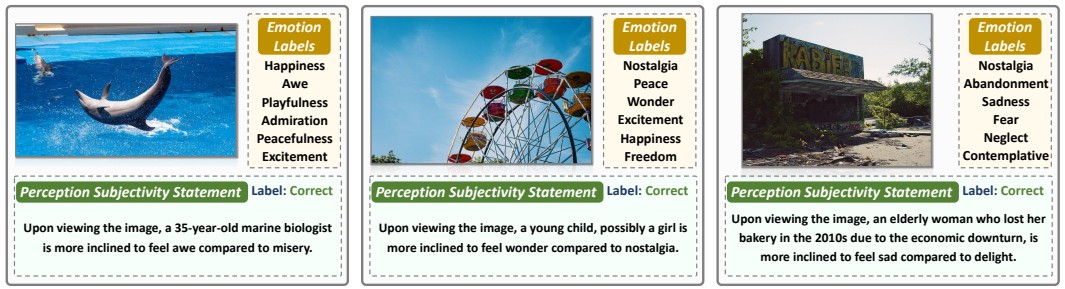

Figure 11: Correct perception subjectivity statements.

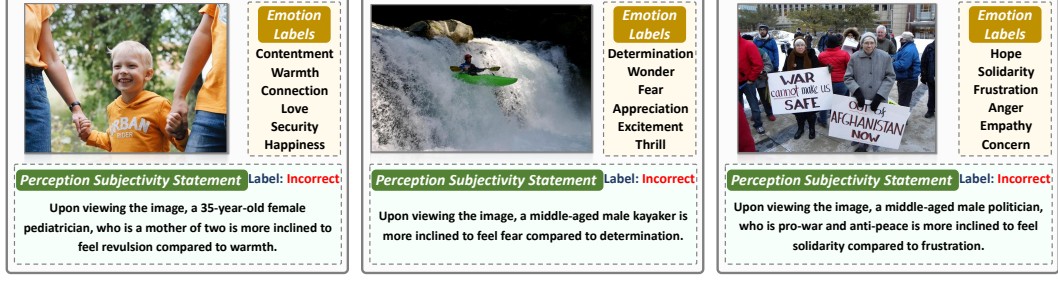

Figure 12: Incorrect perception subjectivity statements.

