# OpenReview forum: "Customizing Visual Emotion Evaluation for MLLMs: An Open-vocabulary, Multifaceted, and Scalable Approach"
_NeurIPS.cc/2025/Conference — Submitted to NeurIPS 2025_

### Official Review · Reviewer_qtK6 · 2025-06-04

**Clarity:** 4
**Significance:** 3
**Originality:** 3
**Rating:** 5
**Confidence:** 4

**Summary:**

The paper is proposing an evaluation benchmark for MLLMs in elicited emotion recognition. To produce the benchmark, images from EmoSet are passed to multiple MLLMs that generate annotations in terms of sentiment, elicited emotion, scene description and perception subjectivity. The captions are then constructed using the annotations and majority voting. Finally, human experts curate a subset of the image-caption sets to verify the caption accuracy and validity.

**Questions:**

The method is clear so not too many questions, but few additions (specifically finetuned and in-context results) would significantly help show the importance of a task specific dataset.

**Ethical Concerns:**

["NO or VERY MINOR ethics concerns only"]

**Final Justification:**

The authors adequately addressed and clarified the concerns raised by me and other reviewers, so I maintain my original positive rating.

**Limitations:**

MLLMs are trained on large amounts of publicly available, often non-curated data. As such, there is always a chance of dataset bias that can be particularly dangerous in sensitive tasks, including emotion recognition. In addition, as this is technically a benchmark for a generative task involving human emotion recognition, an ethics section should be included.

**Paper Formatting Concerns:**

No issues with formatting identified.

**Quality:**

3

**Strengths And Weaknesses:**

The paper is very well written; the authors do a good job explaining the issue of label uncertainty in emotion recognition tasks (for humans and automated systems alike) and propose a reasonable method to address this when evaluating MLLMs. One potential issue with the use of MLLM ensemble and majority voting would be that of dataset bias from the MLLM pre-training affecting the output. It may be worth including all generated captions with the dataset, for validation purposes and so that researchers can investigate label uncertainty/bias further.
One thing that seems inaccurate in the text, is the context of the image being extrinsic or external (mentioned in the intro and line 126 of the method). Not sure is the scene context is extrinsic at all, as that would refer to the context at which the viewer sees the image (eg. on the news or their mobile phone). The scene context is still very much intrinsic to the image, even if sometimes it does not directly relate to the image subject (eg. the EMOTIC examples of Fig 1 (a)).
In terms of experiments, all of the experiments are zero-shot. It would be best to include an example of a fine-tuned model (maybe with LoRA). It would also be good to include some in-context learning experiments from the 3k human annotated subset. Both would be helpful benchmarks.

---

> ### Author Rebuttal · Authors · 2025-07-29
>
> We sincerely appreciate your positive feedback and constructive advice! Below, we present detailed responses to the weaknesses (**W**), questions (**Q**), and limitations (**L**).
>
> >**W1.  One potential issue with the use of MLLM ensemble and majority voting would be that of dataset bias from the MLLM pre-training affecting the output. It may be worth including all generated captions with the dataset, for validation purposes and so that researchers can investigate label uncertainty/bias further.**
>
> Thank you for your suggestion! In the final release, we will provide all generated captions along with other helpful materials to facilitate further research.
>
> >**W2. One thing that seems inaccurate in the text is the context of the image being extrinsic or external (mentioned in the intro and line 126 of the method). Not sure if the scene context is extrinsic at all, as that would refer to the context at which the viewer sees the image (eg. on the news or their mobile phone). The scene context is still very much intrinsic to the image, even if sometimes it does not directly relate to the image subject (eg. the EMOTIC examples of Fig. 1 (a)).**
>
> You are right, this is indeed an imprecise expression. Our original intention was to use "scene context" to represent all contextual elements beyond just the image content. We will carefully proofread the manuscript and replace such oversights with appropriate expressions.
>
> >**W3&Q1. It would be best to include an example of a fine-tuned model (maybe with LoRA). It would also be good to include some in-context learning experiments from the 3k human-annotated subset. Both would be helpful benchmarks.**
>
> For further exploration of the ESJ task, we conduct both fine-tuning and in-context learning (ICL) experiments to provide more comprehensive insights into INSETS-3K. Specifically, we split INSETS-3K into a training set (also used as the support set for ICL, containing 2,708 samples) and a test set (300 samples). The test set is identical to the subset used in Table 4 of the manuscript. We adopt Qwen2.5-VL as the baseline MLLM, and perform LoRA fine-tuning, full-parameter fine-tuning, and ICL with 2, 4, and 8 demonstration samples. The learning rate in fine-tuning is set to 1e-5, the LoRA rank is 16, and ICL demonstrations are randomly retrieved.
>
> | Qwen2.5-VL | Sentiment Polarity | Emotion Interpretation | Scene Context | Perception Subjectivity | Total |
> |---|:---:|:---:|:---:|:---:|:---:|
> | Direct Inference | 64.0 | 81.5 | 83.3 | 68.0 | 76.4 |
> | LoRA Fine-tuning | 79.5 | 84.4 | 84.2 | 69.7 | 80.0 |
> | Full Fine-tuning | 82.1 | 83.5 | 85.8 | 71.1 | 80.7 |
> | 2-shot ICL | 64.6 | 81.7 | 85.5 | 67.1 | 76.7 |
> | 4-shot ICL | 69.2 | 81.8 | 86.3 | 68.4 | 77.9 |
> | 8-shot ICL | 73.6 | 82.8 | 86.8 | 68.8 | 79.1 |
> | Human Average | 92.3 | 90.1 | 95.3 | 89.6 | 91.6 |
>
> As shown in the above table, all these techniques improve MLLM's performance, demonstrating the benefits of task-specific fine-tuning and in-context learning (ICL). The most notable improvement is observed in the sentiment polarity dimension. Interestingly, while this represents a relatively basic task, MLLM's direct inference performance falls short of expectations. This suggests that while the MLLM may possess the inherent capability to perceive overall emotional tone, they tend to confuse between positive, negative, and mixed categories. With a few demonstrations or small-scale fine-tuning, this classification challenge can be effectively overcome. In contrast, MLLM's performance on another challenging dimension, perception subjectivity, only obtained marginal improvements. We attribute this to more fundamental limitations in MLLMs' understanding of subjectivity, and subjectivity-centric training objectives or specialized datasets are warranted.
>
> We will incorporate these results, along with a more comprehensive evaluation of other MLLMs, into the revised manuscript. We hope these can broaden the benchmark's coverage and provide deeper insights into our proposed ESJ task and MLLMs' emotional intelligence.
>
> >**L1. MLLMs are trained on large amounts of publicly available, often non-curated data. As such, there is always a chance of dataset bias that can be particularly dangerous in sensitive tasks, including emotion recognition. In addition, as this is technically a benchmark for a generative task involving human emotion recognition, an ethics section should be included.**
>
> We sincerely appreciate your raising these important concerns regarding potential ethical issues. In line with the suggestion, we will include an ethics section claiming potential risks on:
> 1. Dataset bias and data privacy associated with MLLM pre-training.
> 2. Potentially inaccurate or even misleading information from generative model outputs.
> 3. The inherent subjectivity in emotion perception.
>
> We will also carefully consider other possible ethical risks to mitigate any potential negative societal impacts.

---

> > ### Comment · Reviewer_qtK6 · 2025-08-05
> >
> > Thank you for your detailed rebuttal! The authors have discussed most of my and other reviewers' concerns, which will be taken into consideration for the final review.

---

### Official Review · Reviewer_Tzwg · 2025-06-29

**Clarity:** 3
**Significance:** 1
**Originality:** 2
**Rating:** 2
**Confidence:** 3

**Summary:**

This paper redefines emotion evaluation for MLLMs through a new Emotion Statement Judgment (ESJ) task, where models judge whether a given emotion-related statement about an image is correct. To support this, the authors propose INSETS, a pipeline that automatically extracts open-vocabulary emotion labels using multiple MLLMs and generates emotion-centric statements across four dimensions. They release a large-scale dataset (462K samples) and a 3K subset validated by human annotators for benchmarking.

**Questions:**

I am wondering if showing users the MLLM prediction and asking them to judge this evaluation is less reliable that having the human label the image first and then compare that to the MLLM result.

**Ethical Concerns:**

["NO or VERY MINOR ethics concerns only"]

**Final Justification:**

I have read the rebuttal and the other reviewers’ comments. While I appreciate the effort put into this work, I still believe that it is primarily a dataset contribution whose reproduction is not easily scalable due to the human annotation effort required. As such, I feel it would be better suited for the Datasets and Benchmarks track rather than the general NeurIPS submission track. My score remains a reject.

**Limitations:**

Yes

**Quality:**

2

**Strengths And Weaknesses:**

### Strengths
- The paper introduces a novel reformulation of emotion evaluation in MLLMs via the ESJ task.
- The proposed INSETS pipeline is described in thorough detail, covering open-vocabulary emotion extraction, taxonomy mapping, and multi-perspective statement construction.
- The authors evaluate a broad suite of MLLMs across the ESJ (3k) benchmark, and release a human-verified subset for standardized comparison.

### Weaknesses
- The dataset relies on majority voting between existing MLLMs to generate emotion labels and statements. However, these very models are shown by the authors to have a large gap compared to humans.
- The evaluation of a highly subjective task like emotion judgment lacks rigor without expert human annotations or deeper analysis of inter-subject variability. The annotators’ qualifications and agreement metrics are also not discussed.
- As it stands, the work primarily offers a pipeline and benchmarking framework that is difficult to subjectively verify.
- Given these limitations, particularly around label reliability and subjectivity, the paper falls short of the bar for acceptance in its current form.

---

> ### Author Rebuttal · Authors · 2025-07-29
>
> We sincerely appreciate your thorough review and rigorous suggestions! Below, we present detailed responses to the weaknesses (**W**) and questions (**Q**).
>
> >**W1. The dataset relies on majority voting between existing MLLMs to generate emotion labels and statements. However, these models are shown to have a large gap between humans.**
>
> We have carefully considered this aspect and placed great emphasis on ensuring reliability during benchmark construction. Instead of directly applying MLLM-generated statements, we primarily employ MLLMs to produce preliminary results, such as open-vocabulary emotion candidates and prototype statements. Building upon these outputs, **we assign appropriate open-vocabulary emotions based on well-established emotion theories and design strategic rules to construct image-statement pairs with minimal ambiguity**. Most importantly, **we introduce necessary human intervention for correction and verification to maximize the reliability of the constructed benchmark**, particularly for INSETS-3K. Below, we elaborate on our step-by-step designs for mitigating inaccuracies inherent to MLLMs.
>
> In the open-vocabulary emotion tagging stage, after employing MLLMs to analyze the emotions of each image, we adopt a coarse-to-fine approach to attach them to Parrott's hierarchical model. First, GPT-4 is employed to establish coarse relationships. Then, **a psychology postgraduate is hired to refine these relationships from a professional perspective**, ensuring the reliability of the obtained Parrott-based open-vocabulary hierarchical model. Based on this model, we apply majority voting across MLLMs to select emotions that receive consistent recognition. While we acknowledge that the annotations may not fully align with certain individuals' perceptions, the inherent subjectivity of emotion precludes a definitive ground truth. Thereby, we do not proceed with manual filtering at this stage, aiming to preserve the potential diversity of subsequent statements.
>
> In the statement construction stage, we first ask MLLMs to produce prototype statements and then construct statements under carefully designed rules. **These rules are elaborated to simultaneously minimize potential ambiguities in statements while preserving appropriate task difficulty.** To further ensure label reliability in INSETS-3K, we incorporate a manual verification step. Specifically, **we engage five graduate students to evaluate the accuracy of automatically assigned labels**. Each image-statement pair is retained only if at least four annotators reach a consensus on its label. Through verification, 90.6% of automatically assigned labels are validated as accurate, 6.9% are identified as inaccurate, and 2.5% are discarded due to a lack of consensus. This process not only guarantees the high quality of final labels but also substantiates the inherent reliability of our automated pipeline.
>
> Furthermore, the human evaluation results in Table 4 further confirm the high quality of the benchmark. On the 300-sample subset of INSETS-3K, human participants achieve an average accuracy of 91.6% and a top accuracy of 95.2%, demonstrating that **the final benchmark shows strong alignment with human perception**. This result verifies that the benchmark's quality is unaffected by inconsistencies in MLLMs.
>
> >**W2. The evaluation of emotion judgment lacks rigor without expert human annotations or deeper analysis of the annotation process.**
>
> As mentioned in lines 163–164 and 227–228 of the manuscript, and further discussed in our response to **W1**, we introduce necessary human annotations to ensure the reliability of the benchmark, particularly for INSETS-3K. Below, we provide additional details regarding **annotator qualifications, statistical details, and annotation analysis.** To enhance rigor, we will include this information in the revised manuscript and release the original annotation materials.
>
> Specifically, in attaching open-vocabulary emotions to Parrott's model, we engage a psychology postgraduate to refine the coarse attachments initially assigned by GPT-4. **This annotator has formal training in affective science and professional expertise in emotion theories.** The process takes approximately 15 hours, during which 18.4% of the 2.2k attachment relationships are manually revised.
>
> Following the automated construction of INSETS-462K, we invite five graduate students to perform manual verification on its subset, INSETS-3K. Their research direction mainly lies in affective computing; aged between 22-27; and have received task-specific guidance. **For each image-statement pair, its label is confirmed only if at least four annotators agree.** This process ensures that the final labels align with humans' perception. The table below reports the proportion of labels manually verified as accurate for each task dimension.
>
> |Accuracy|Sentiment Polarity|Emotion Interpretation|Scene Context|Perception Subjectivity|Total|
> |-|:-:|:-:|:-:|:-:|:-:|
> |Assigned as Correct|94.9|86.2|94.6|87.5|89.7|
> |Assigned as Incorrect|93.4|92.0|93.4|88.0|91.5|
>
> From the results, all dimensions demonstrate high construction accuracy. Among these, statements on emotion interpretation and perception subjectivity exhibit slightly lower accuracy. Within the emotion interpretation dimension, only 86.2% statements assigned as correct are verified as accurate. According to the pipeline, they are constructed by directly combining emotional states with the corresponding MLLM-generated prototype interpretations, which are more vulnerable to inaccuracies in MLLMs.
>
> For the perception subjectivity dimension, all statements show construction accuracy lower than 88.0%. In constructing these statements, MLLMs are instructed to generate prototype characters by describing who might experience the target emotion when viewing the image. Our pipeline then randomly samples an emotion from the opposite polarity spectrum as what this character would be less likely to feel. We hypothesize that this approach may be affected by MLLMs' limitations in comprehending subjectivity, potentially resulting in overly generic prototype characters and consequently ambiguous statements.
>
> | |Sentiment Polarity|Emotion Interpretation|Scene Context|Perception Subjectivity|Total|
> |-|:-:|:-:|:-:|:-:|:-:|
> |5/5|61.0|42.5|78.1|44.0|54.0|
> |4/5|33.2|46.6|15.9|43.7|36.6|
> |3/5|1.3|1.3|1.3|1.7|1.4|
> |2/5|1.0|0.7|0.4|2.4|1.1|
> |1/5|2.5|3.6|1.9|3.9|3.1|
> |0/5|1.0|5.3|2.4|4.3|3.8|
> |Kappa|0.68|0.51|0.81|0.52|0.61|
>
> In the table above, we report the distribution of annotator agreement and the Fleiss’ Kappa scores. **The Kappa statistic quantifies inter-annotator agreement**. These results reveal a pattern consistent with previous analysis. Annotators demonstrate overall high agreement, with only marginally lower consistency in emotion interpretation and perception subjectivity dimensions.
>
> **Importantly, these relatively infrequent misassignments are effectively identified and corrected during human verification, ensuring the final quality and reliability of INSETS-3K**.
>
> >**W3&W4. The work primarily offers a pipeline and benchmarking framework that is difficult to subjectively verify. There are limitations around label reliability and subjectivity.**
>
> We hope our responses to **W1** and **W2** can address your concerns regarding the verification of our benchmark construction and its reliability.
>
> We fully acknowledge that emotion perception is inherently subjective, which is a key motivation behind proposing the ESJ task. This task formulation helps minimize ambiguity in evaluation while maintaining strong extensibility in both breadth and depth. During the manual verification process of INSETS-3K, the overall Fleiss’ Kappa reaches 0.61, indicating **substantial inter-annotator agreement**. This demonstrates that the ESJ task design, combined with the INSETS pipeline, effectively reduces subjectivity in emotion evaluation, establishing a reliable foundation for visual emotion evaluation, even in the absence of fully objective ground truth.
>
> >**Q1. I am wondering if showing users the MLLM prediction and asking them to judge this evaluation is less reliable than having the human label the image first and then comparing that to the MLLM result.**
>
> We assume your concern might be whether directly evaluating MLLM-generated outputs against human-annotated captions is more reliable. In that case, please refer to our response to **W1 of Reviewer Hn9F** for a detailed explanation.
>
> Alternatively, if your concern lies in the replacement of the manual verification process in INSETS-3K with direct human evaluations of statement accuracy, we also appreciate this important perspective. It helps validate whether the MLLM-assigned labels introduce prior biases that influence human judgments. Due to time and labor constraints, we conduct a toy experiment on 100 samples from the pre-filtered INSETS-3K to explore this issue. To avoid potential conflicts between human decisions and MLLM-assigned labels, we adopt a simplified setting. Five annotators (graduate students aged 23–27) independently assess each statement's correctness, and final labels are determined using the same voting rules as before.
>
> Only two previously labeled "incorrect" statements are rejudged as ambiguous, while all others remain consistent, affecting just 2% of the samples. This suggests the discrepancy between annotation methods is minor and is unlikely to influence our conclusions. We attribute this to ESJ's task format, which effectively minimizes ambiguity. However, due to the inherent subjectivity of emotion perception, it remains challenging to determine which method yields more accurate labels. We plan to investigate this annotation discrepancy further in future work.
>
> We apologize if our interpretation still misses the essence of your concern. Please do not hesitate to clarify further, and we would be more than happy to address it.

---

### Official Review · Reviewer_Hn9F · 2025-07-03

**Clarity:** 3
**Significance:** 2
**Originality:** 2
**Rating:** 4
**Confidence:** 2

**Summary:**

The paper introduces a new emotion evaluation benchmark for MLLMs. They propose a new task called ESJ which classifies whether an emotional sentence is correct relative to the input image or not. For that they create an automatic annotation pipeline to create sentences that cover aspects of sentiment polarity, emotion interpretation, scene context, and perception subjectivity. They benchmark existing MLLMs on the new ESJ task underscoring shortcomings of existing models.

**Questions:**

Please refer to weakness

**Ethical Concerns:**

["NO or VERY MINOR ethics concerns only"]

**Final Justification:**

The author has already addressed my concerns, so I have decided to raise my score from 3 to 4. Meanwhile, regarding the question about how to evaluate the emotional statements generated by MLLMs themselves, I recommend that the author propose their own evaluation model.

**Limitations:**

yes

**Paper Formatting Concerns:**

There are no paper formatting concerns.

**Quality:**

3

**Strengths And Weaknesses:**

Strengths:
1.This paper is the first to identify key limitations in existing visual emotion evaluation frameworks for MLLMs, and addresses these shortcomings through the introduction of a novel task—Emotional Statement Judgment (ESJ).

2.The authors propose the Intelligent Visual Emotion Tagger and Statement Constructor (INSETS) pipeline, which enables efficient image annotation with diverse open-vocabulary emotion labels and generates multifaceted, emotion-centered statements while minimizing manual effort.

Weaknesses:
1.The INSETS pipeline offers only limited support for the broader task of evaluating image-based emotions in MLLMs. Framing the task as a binary judgment of whether a statement matches an image appears somewhat simplistic. A more impactful direction would be to explore how to evaluate the emotional statements generated by MLLMs themselves, rather than having MLLMs assess the correctness of predefined emotional statements.

2.The paper’s contributions are relatively modest, focusing mainly on the proposed benchmark and evaluation framework. Research on the ESJ task remains limited.

3.Too many acronums make the paper hard to read.

---

> ### Author Rebuttal · Authors · 2025-07-29
>
> We sincerely appreciate your thoughtful effort in conducting a valuable review! Below, we present detailed responses to the weaknesses (**W**) and questions (**Q**).
>
> >**W1. Framing the task as a binary judgment appears somewhat simplistic, offering limited support for the broader task of evaluating image-based emotions in MLLMs. A more impactful direction would be to explore how to evaluate the emotional statements generated by MLLMs themselves.**
>
> Our primary objective in introducing the binary judgment task is to **avoid ambiguity during the evaluation**. Visual emotion perception is a highly subjective task, where each question permits multiple reasonable answers from diverse perspectives. However, current evaluation tasks, such as emotion classification or interpretation, provide fixed annotations and ignore other plausible answers. This makes them insufficient for accurate emotion evaluation, especially for MLLMs that have not undergone task-specific fine-tuning. In contrast, **the binary judgment task is designed to reduce the space for multiple plausible responses by focusing on the correctness of a single emotional statement.** When combined with the manual curation process of INSETS-3K, it significantly improves the accuracy and reliability of MLLMs' visual emotion evaluation.
>
> **In terms of evaluation breadth and depth, our proposed Emotion Statement Judgment task also offers exceptional extensibility**. Through carefully designed statements, the INSETS benchmark incorporates not only mainstream evaluation dimensions (sentiment polarity and emotion interpretation) but also extends to previously underexplored dimensions: scene content and perception subjectivity. These advantages introduced by the task type are also recognized by **Reviewer JSNz**.
>
> Your suggestion regarding the direct evaluation of MLLM-generated statements is highly insightful. While we fully agree that this approach would provide a comprehensive evaluation of MLLMs' emotional expression capabilities, several significant challenges currently prevent its practical implementation. To assess the quality of MLLM-generated statements, there are two main branches: comparison with fixed annotations or employing more advanced MLLMs as the judger (also known as LLM-as-judge).
>
> As discussed in the first paragraph, the first branch risks unfairly penalizing plausible responses due to the open-ended nature of emotion interpretation, suffering from non-negligible evaluation inaccuracies. As for the second branch, a judger model equipped with superior emotional intelligence is required. However, as demonstrated in Table 3 of the manuscript, even state-of-the-art MLLMs like GPT-4o exhibit non-negligible performance gaps compared to humans on INSETS-3K. This suggests that currently, no model may be truly qualified for the judger role, rendering both evaluation approaches for MLLM-generated statements problematic in practice.
>
> Moreover, the scarcity of pioneering benchmarks leads to inadequate consensus about MLLMs' emotional intelligence. It drives us to focus on more fundamental emotion evaluation, which we hope can serve as both a basis and an indicator for future research. We anticipate that through optimizing performance based on INSETS benchmarks, the LLM-as-judge approach for directly evaluating MLLM statements may become feasible in the near future.
>
>
> >**W2. Contributions are relatively limited, focusing mainly on the proposed benchmark and evaluation framework. Research on the ESJ task remains limited.**
>
> We respectfully disagree with the view that our contribution is limited. Equipping MLLMs with emotional intelligence is a crucial step toward artificial general intelligence. However, current efforts in evaluating MLLMs' visual emotion perception still heavily rely on benchmarks originally designed for small fine-tuned models. While these benchmarks have made significant progress in annotation granularity, quality, and even data scale, they remain insufficient for evaluating MLLMs due to several limitations: the oversight of plausible responses, limited emotional taxonomies, the neglect of extra-visual factors, and the labor-intensive nature of annotations (detailedly explained in lines 39–55 of the manuscript).
>
> Based on this observation, we argue that an evaluation framework and benchmarks tailored for MLLMs would offer a more accurate understanding of their emotion perception capabilities. Such advancements would also lay a solid foundation for further progress in both MLLMs and affective computing.
> Specifically, we categorize our main contributions into threefold:
>
> 1. We identify the limitations of existing visual emotion evaluation methods for MLLMs and customize a fundamental yet pioneering evaluation task for MLLMs. This task effectively mitigates ambiguity in open-ended questions while maintaining extensibility in evaluation breadth and depth, as also elaborated in our response to **W1**.
>
> 2. We develop the INSETS pipeline, which achieves reliable open-vocabulary emotion tagging and emotion statement construction with minimal human effort. Through manual verification and data analysis (Section 4.4, Table 2, and Figure 4 in the manuscript) of INSETS-3K, we demonstrate that the pipeline assigns an average of 5.2 open-vocabulary emotions per image and achieves 90.6% accuracy in label assignment for image-statement pairs. Thanks to its high automation and independence from external image information, this pipeline can be applied to images from any domain, offering potential value for both benchmark diversification and data augmentation.
>
> 3. Leveraging the obtained INSETS-3K benchmark, we conduct a systematic evaluation of popular MLLMs across four key dimensions (sentiment polarity, emotion interpretation, scene context, and perception subjectivity). Our analysis reveals non-negligible performance gaps between MLLMs and humans, particularly in perception subjectivity. These findings provide valuable insights for future research directions in both affective computing and MLLMs.
>
> Regarding further research on the ESJ task, our primary objective is to introduce this task for evaluation and establish a benchmark for general-purpose MLLMs, rather than proposing specific model improvements for this task. However, we fully agree with you that further exploration could yield valuable insights. In response to this consideration, we take suggestions from **Reviewer qtK6** and conduct both fine-tuning and in-context learning (ICL) experiments to provide deeper analysis (most of the following content is borrowed from our response to **W3&Q1 of Reviewer qtK6**).
>
> Specifically, we split INSETS-3K into a training set (also used as the support set for ICL, containing 2,708 samples) and a test set (300 samples). The test set is identical to the subset used in Table 4 of the manuscript. We adopt Qwen2.5-VL as the baseline MLLM, and perform LoRA fine-tuning, full-parameter fine-tuning, and ICL with 2, 4, and 8 demonstration samples. The learning rate in fine-tuning is set to 1e-5, the LoRA rank is 16, and ICL demonstrations are randomly retrieved.
>
> | Qwen2.5-VL | Sentiment Polarity | Emotion Interpretation | Scene Context | Perception Subjectivity | Total |
> |---|:---:|:---:|:---:|:---:|:---:|
> | Direct Inference | 64.0 | 81.5 | 83.3 | 68.0 | 76.4 |
> | LoRA Fine-tuning | 79.5 | 84.4 | 84.2 | 69.7 | 80.0 |
> | Full Fine-tuning | 82.1 | 83.5 | 85.8 | 71.1 | 80.7 |
> | 2-shot ICL | 64.6 | 81.7 | 85.5 | 67.1 | 76.7 |
> | 4-shot ICL | 69.2 | 81.8 | 86.3 | 68.4 | 77.9 |
> | 8-shot ICL | 73.6 | 82.8 | 86.8 | 68.8 | 79.1 |
> | Human Average | 92.3 | 90.1 | 95.3 | 89.6 | 91.6 |
>
> As shown in the above table, all these techniques improve MLLM's performance, demonstrating the benefits of task-specific fine-tuning and in-context learning (ICL). The most notable improvement is observed in the sentiment polarity dimension. Interestingly, while this represents a relatively basic task, MLLM's direct inference performance falls short of expectations. This suggests that while the MLLM may possess the inherent capability to perceive overall emotional tone, they tend to confuse between positive, negative, and mixed categories. With a few demonstrations or small-scale fine-tuning, this classification challenge can be effectively overcome. In contrast, MLLM's performance on another challenging dimension, perception subjectivity, only obtained marginal improvements. We attribute this to more fundamental limitations in MLLM's understanding of subjectivity, and subjectivity-centric training objectives or specialized datasets are warranted.
>
> We will incorporate these results, along with a more comprehensive evaluation of other MLLMs, into the revised manuscript. We hope these can broaden the benchmark's coverage and provide deeper insights into our proposed ESJ task and MLLMs' emotional intelligence.
>
> >**W3. Too many acronyms make the paper hard to read.**
>
> Thank you for pointing it out! We will carefully examine the necessity of acronyms and reduce their number to an appropriate level.

---

> > ### Comment · Reviewer_Hn9F · 2025-08-04
> >
> > The author has already addressed my concerns, so I have decided to raise my score from 3 to 4. Meanwhile, regarding the question about how to evaluate the emotional statements generated by MLLMs themselves, I recommend that the author propose their own evaluation model.

---

> > > ### Author Response · Authors · 2025-08-05
> > >
> > > Thank you again for your valuable time! We will carefully consider your recommendation in our future work.

---

### Official Review · Reviewer_JSNz · 2025-07-05

**Clarity:** 4
**Significance:** 3
**Originality:** 3
**Rating:** 5
**Confidence:** 4

**Summary:**

The paper proposes an evaluation framework for subjective, taxonomy-rich task such as the emotion understanding task from visual context. Their core argument that conventional approaches like Emotion Classification and Emotion Interpretation are incompatible with modern MLLMs due to "limited emotional taxonomies" and "rigid evaluation metrics" is central to their thesis and is a salient one. Their proposed four dimensions of the "ESJ" task: Sentiment Polarity, Emotion Interpretation, Scene Context, and Perception Subjectivity is comprehensive and addresses various aspects of emotion perception. Benchmark dataset (with a human-validated subset) and results are compellinh.

**Questions:**

My question concerns the ground-truth generation, which effectively defines your scoring function. The INSETS pipeline anchors open-vocabulary terms to the fixed hierarchy of Parrott's model to determine the 'correctness' of a statement.

In this process, did you observe any inherent contradictions or challenging edge cases? For example, were there open-vocabulary emotions that did not fit into clean categorization within Parrott's rigid structure, or instances where an automatically generated 'incorrect' statement was, in fact, subjectively plausible? Given MLLMs can be inherently inconsistent?

**Ethical Concerns:**

["NO or VERY MINOR ethics concerns only"]

**Limitations:**

No major limitations

**Quality:**

3

**Strengths And Weaknesses:**

Strengths
1. The paper does a good job at framing the central problem: the core mismatch between existing visual emotion benchmarks and the capabilities of modern MLLMs. The introductory examples (Fig. 1 a,b) are particularly effective, clearly showing how plausible MLLM responses can be unfairly penalized by rigid, closed-vocabulary, or fixed ground-truth evaluation schemes.
2. The solution framed as "Emotion Statement Judgment" instead of classification or generation allows the statements themselves to be complex and nuanced, but the eval metric to be less ambiguous relying on binary yes/no answers.
3. Perhaps the most important, The evaluation of current MLLMs (Table 3, 4) shows that performance gap between top MLLMs like GPT-4o and humans provides a clear signal to the community that visual emotional "intelligence" is still an open problem. I find this insightful

Weaknesses:
1. Minor: The hierarchical model of the Parrott's theory doesn't allow for continuous mixed emotions (think arousal-valence model) but the "perception subjectivity" dimension address this somewhat

---

> ### Author Rebuttal · Authors · 2025-07-29
>
> We sincerely appreciate your positive feedback and helpful comments! Below, we present detailed responses to the weaknesses (**W**) and questions (**Q**).
>
> >**W1. The hierarchical model of Parrott's theory doesn't allow for continuous mixed emotions.**
>
> Indeed, dimensional emotion space (DES) models (such as the arousal-valence model) provide a precise way to characterize psychological states using continuous coordinates. However, for humans' daily communication and popular MLLMs, emotions are more commonly expressed in natural language form, where they inevitably appear as discrete points within the DES. In this context, Parrott's theory—with its 113 finely grained tertiary emotions—enables relatively accurate emotion expressions. By augmenting it with open-vocabulary leaves, we further refine the granularity for MLLMs' visual emotion evaluation within the natural language dimension.
> On the other hand, emotion evaluation based on DES models is a promising direction, and we plan to explore this further in future work.
>
> >**Q1. Inherent contradictions and challenging edge cases in the benchmark construction process.**
>
> The process of constructing the benchmark involves several challenges, which we group into three major aspects.
>
> **First**, when mapping open-vocabulary emotions to Parrott's model, certain complex or mixed emotions (e.g., "peaceful alertness" or "solitary introspection") prove difficult to assign to appropriate anchors. Initially, GPT-4 is employed for coarse-grained attachment, but approximately 10.7% of open-vocabulary emotions are flagged as "not applicable" to any of the tertiary emotions. To refine the mapping, we hire a psychology postgraduate to reassess these ambiguous cases as well as optimize other rough attachment relations from a professional perspective. As a result,  we successfully attach the vast majority of open-vocabulary emotions, with only 1.2% remaining unresolvable and subsequently discarded. For instance, "peaceful alertness" and "solitary introspection" are attached to "alarm" and "melancholy" respectively.
>
> **Second**, in emotional expression, there is no golden standard to clearly separate error from subjectivity. Therefore, we place great emphasis on minimizing potential ambiguities during the automated construction. To further ensure reliability, we also engage five graduate students to conduct manual verification of the INSETS-3K dataset. Each image-statement pair is retained only if at least four annotators reach a consensus on its label. Despite this process, approximately 2.5% of the pairs remain in disagreement and are subsequently excluded from INSETS-3K.
>
> **Third**, as demonstrated in Table 3 of the manuscript, the MLLMs involved in benchmark construction inherently exhibit certain inaccuracies. To mitigate this impact, our INSETS pipeline incorporates two key designs: (1) an open-vocabulary emotion tagging stage based on majority voting, and (2) a strategic statement construction stage. Through manual verification, we confirm that the labels of 90.6% statements in INSETS-3K receive correct label assignments. When combined with the diversity analysis presented in Table 2 and Figure 4 of the manuscript, these results demonstrate that our proposed pipeline effectively constructs high-quality samples while substantially reducing the impact of MLLMs' inconsistencies. Benefiting from the subsequent manual verification, the final INSETS-3K benchmark is completely free from such effects, ensuring both high quality and reliability.

---

### Decision · Program_Chairs · 2025-09-17

**Decision:**

Reject

**Comment:**

This paper introduces a novel framework for evaluating the ability of MLLMs to perceive emotions in images. The proposed evaluation is based on the Emotion Statement Judgment (EMJ) task, which assesses model performance across four types of emotion-centric statements: sentiment polarity, emotion interpretation, scene context, and perception subjectivity. The paper also presents an evaluation of several existing MLLMs using the proposed framework and benchmark.

All the reviewers value the motivation of the paper and how the paper frames the challenges associated with evaluating emotion perception in images. However, the paper has received mixed scores. During the first round of reviews, the reviewers identified the following key weaknesses: (1) The paper’s contributions were considered relatively modest, focusing primarily on the introduction of a new benchmark and evaluation framework; (2) Concerns were raised about the ground truth generation process. Specifically, the new evaluation benchmark derives emotion labels through majority voting among existing MLLMs, while the results of the paper show that MLLMs exhibit a significant gap compared to humans in emotion perception from images. This also raised questions about the potential introduction of bias through the use of MLLMs for annotation; (3) Although the paper claims that the proposed evaluation framework is scalable, it still depends heavily on human annotation, which challenges the claim of scalability.

The Reviewer-Author discussion has been active and constructive, generating several ideas for addressing the weaknesses of the paper. However, following further deliberation during the Reviewer-AC discussion period, both the reviewers and AC still identified significant concerns that suggest the paper is not yet ready for acceptance. In particular, two of the reviewers still consider the contribution to be limited. It was also noted that the paper’s main contribution (the new evaluation framework based on specific evaluation dimensions) lacks grounding in solid or well-established psychological research. We recommend the authors address this issue and the other suggestions provided by the reviewers to strengthen the work and improve its overall impact.